# Oligometastatic Disease: When Stage IV Breast Cancer Could Be “Cured”

**DOI:** 10.3390/cancers14215229

**Published:** 2022-10-25

**Authors:** Maria Gion, Cristina Saavedra, Jose Perez-Garcia, Javier Cortes

**Affiliations:** 1Medical Oncology Department, Ramón y Cajal University Hospital, 28034 Madrid, Spain; 2Instituto Ramón Y Cajal de Investigación Sanitaria (IRYCIS), 28034 Madrid, Spain; 3International Breast Cancer Center (IBCC), Pangaea Oncology, Quironsalud Group, 08017 Barcelona, Spain; 4Medical Scientia Innovation Research (MedSIR), 08018 Barcelona, Spain; 5Faculty of Biomedical and Health Sciences, Department of Medicine, Universidad Europea de Madrid, 28670 Madrid, Spain

**Keywords:** metastases, breast cancer, oligometastatic disease, radiotherapy, surgery

## Abstract

**Simple Summary:**

Breast cancers diagnosed at an advanced stage are often incurable, and systemic strategies such as chemotherapy, hormonotherapy, immunotherapy, and/or targeted therapy are the basis of treatment. Imaging tests are always required to determine the extent of the disease, and in selected cases with few metastatic lesions and organs involved (oligometastatic disease), radical treatment with surgery or radiotherapy may be considered to improve prognosis. In that situation, although it is basically impossible to know whether it is a curable disease or not, more and more patients achieve the status of no long-term evidence of disease. More prospective studies are needed to assess which patients may benefit.

**Abstract:**

Although metastatic breast cancer remains an incurable disease, there are patients with a limited number of metastatic lesions that, in addition to systemic therapy, can be treated with “radical therapy” and sometimes reach the status of no long-term evidence of disease. Whether or not these patients can be considered cured is still a matter of debate. Unfortunately, the definition of the oligometastatic disease remains unclear, and it can occur with multiple different presentations. The absence of remarkable biomarkers, the difficulty in designing the appropriate clinical trials, and the failure to offer this group of patients radical approaches in advanced-stage clinical trials are just some of the current problems that we face in treating patients with oligometastatic breast cancer. Although most of the data come from retrospective studies and do not use the same definition of “oligometastatic disease,” here we review the main studies exploring the role of surgery or radiotherapy in patients with the oligometastatic disease and the different results. Some, but not all, studies have shown an increase in survival when surgery and/or radiotherapy were performed for oligometastatic disease. However, better clinical trial designs are needed to confirm the role of “aggressive” approaches for patients with breast cancer and oligometastatic disease.

## 1. Introduction

Metastatic breast cancer (MBC) is generally considered an incurable disease. However, in recent years there has been a significant increase in survival. At least two key aspects are involved in this improvement. On the one hand, advances in cancer treatments have clearly improved disease control and overall survival; on the other hand, better imaging tools are beginning to be used to diagnose metastatic breast cancer with less disease. It is anticipated that in the upcoming years, with the wider use of more sensitive and new potential techniques such as liquid biopsy, many more patients will be diagnosed with micrometastatic and/or oligometastatic disease.

There is no standard definition of oligometastatic disease. In fact, the most accepted definition includes when a maximum of five metastatic lesions are involved. This definition does not include the number of affected organs (Figure 1a,b). Others have proposed oligometastatic disease when a radical approach to all metastatic lesions could potentially be performed (Figure 1c) [1].

The oligometastatic disease accounts for approximately 1–10% of all MBCs [2]. There is a lack of well-designed prospective randomized clinical trials for this group of patients. Also, although they may be included in advanced-stage clinical studies, radical approaches (including surgery or radiation therapy) are not allowed. No study, to the best of our knowledge, has been designed to recruit patients with oligometastatic disease, optimizing not only the best systemic treatment according to the different subtypes of breast cancer but also allowing the optimal radiation/surgical approach at the best time. It is true that the current literature seems to show that a significant number of these patients might have very long overall survival. Would it be possible that some of them can even be cured? Although it is basically impossible to know if it is a curable disease or not, more and more patients achieve the status of long-term no-evidence of disease, basically when they are treated not only with systemic therapy but also with local treatments, including surgery or radiotherapy.

Through this manuscript, we review the main studies exploring the role of surgery or radiotherapy in patients with the oligometastatic disease and the different results. Some, but not all, studies have shown an increase in survival when surgery and/or radiotherapy are performed for oligometastatic disease. However, as stated previously, better clinical trial designs are needed to confirm the role of “aggressive” approaches for patients with breast cancer and oligometastatic disease.

## 2. Locoregional Treatment

When patients are diagnosed with de novo metastatic breast cancer, systemic-based treatment is usually the preferred approach. However, there is a clear debate about the role of primary surgery. Excluding patients with ulceration or pain in the breast/axilla, there is no consensus on the optimal management of the locoregional disease. Different biological hypotheses have been postulated to justify or not the surgery of the primary tumor. One of the most accepted refers to the immunomodulatory effect after the removal of the primary tumor. In fact, nephrectomy has been the optimal approach to treat patients with stage IV renal cell carcinoma for many years [3]. On the other hand, it has also been postulated that surgery in patients with the de novo metastatic disease may carry a higher risk of progression due to increased growth factors and possible immunosuppression [4,5].

If locoregional treatment is considered, optimal management is also controversial. If the breast tumor can be removed without positive margins, conservative surgery is generally preferred. Regarding the axilla, different possibilities should be considered: (i) axillary lymphadenectomy, (ii) sentinel node biopsy and (iii) wait-and-see attitude. This decision can also be considered based on the status of the axilla at baseline and the activity of the systemic treatment.

Retrospective studies evaluating the role of locoregional management in patients with MBC have shown conflicting results. These data should be taken with caution because most of these studies have selection biases (for example, younger patients in the surgery cohorts). Furthermore, how many patients had the oligometastatic disease was not adequately reported in many of these studies. Lastly, many of the latest advances in systemic cancer treatments that increase survival were not part of these studies, so the data could not be extrapolated to the present [6].

A meta-analysis that included 10 studies (nine retrospective studies and one case-control study) analyzed the 3-year survival in 28,693 patients with stage IV breast cancer treated with either systemic therapy alone or systemic therapy plus primary surgery [7]. 52.8% of patients underwent surgery for the primary tumor, showing a superior 3-year survival (OR 2.32, 95% CI 2.08–2.6, *p* < 0.01). Twenty-two percent of patients who received only systemic treatment were alive, compared to 40% who were also treated with surgery. No differences in survival were found between patients treated with mastectomy and breast-conserving surgery. Patients with oligometastatic disease seem to benefit more from surgery [8]. One of the major limitations of this analysis relates to the lack of knowledge of HER2, as well as management with appropriate anti-HER2 therapies.

Cleopatra is a randomized, placebo-controlled, phase III trial in patients with HER2-positive advanced or metastatic breast cancer who were randomized to receive docetaxel, trastuzumab and either placebo or pertuzumab in the first-line setting. A significant improvement in overall survival (OS) improvement was observed among those patients treated with double blockade [9]. Of interest, an ad-hoc analysis from this study explored the role of primary surgery. The 3-year OS was better in the surgery group (74.1% vs. 53.3%). The median OS in patients treated with systemic therapy alone was 39.8 months and was not reached in patients treated with surgery [10]. Unfortunately, it is unknown how many of these patients had the oligometastatic disease.

Recently, some prospective, randomized trials have explored the role of breast surgery in patients with MBC showing conflicting results. In the Turkish phase III study, Soran et al. randomized patients to be treated with breast surgery followed by systemic treatment versus systemic treatment alone [11]. At 10 years of follow-up, 19% (95% CI 13–28%) of patients were alive in the surgical cohort versus 5% (95% CI 2–12%) in the systemic treatment-alone cohort [12]. This benefit was more significant in the group of patients who had Estrogen Receptor (ER)/Progesterone Receptor (PR) positive and HER2-positive tumors and for patients with solitary bone metastases. It should be noted that the subgroups were not well balanced (more ER-positive, younger patients and few metastases in the surgical group).

In a second study, Badwe et al. randomized patients with *de novo* metastases who had responded to first-line anthracycline-based therapy to local treatment versus no local treatment [13]. No difference in OS was observed between both groups of patients (median OS, 19.2 months vs. [95% CI 15.98–22.46] vs. 20.5 months [16.96–23.98]; HR 1.04, 95% CI 0.81–1.34; *p* = 0.79). Local treatment improved locoregional progression-free survival (PFS) but led to worse distant disease-free survival. Patients with the oligometastatic disease also did not show any benefit from local therapy. However, it has not been reported whether these patients were also treated with an aggressive approach to metastatic sites.

The phase III ABCSG-28 POSYTIVE trial compared primary surgery followed by systemic therapy versus systemic treatment alone for patients with stage IV breast cancer with de novo metastases [14]. This trial had to be stopped due to poor recruitment. Nevertheless, although there was no benefit in terms of OS, there was a numerically worse median OS for those patients treated with local surgery (34.6 months in the group of patients treated with primary surgery and 54.8 months in the group without surgery) (HR favored patients with no surgery 0.691; 95% CI 0.358–1.333; *p* = 0.267).

Although the comparison of different clinical trials is not precise, it should be noted that in the Soran trial, more patients had bone-only involvement (47% vs. 28% in the Badwe study and 27% in the POSYTIVE trial). This could be associated with the benefit seen in this study compared to the other two studies.

The ECOG-ACRIN E2108 study randomized 390 patients with stage IV breast cancer to receive or not receive locoregional treatment (surgery or radiation therapy). Patients had to have the nonprogressive disease after 4–8 months of systemic therapy. This trial showed no difference between the two groups in OS at 3 years (68.4% locoregional treatment vs. 67.9%, *p* = 0.63, HR: 1.09, 90% CI: 0.80, 1.49). Of interest, 20% of patients randomized to the locoregional approach in this trial did not have negative margins at surgery [15].

Disease burden has been correlated with worse outcomes in many clinical trials [7]. Some investigators have questioned the role of an aggressive locoregional approach for only those patients with more indolent disease. The BOMET MF14-01 study randomized patients with de novo stage IV breast cancer with only bone metastases to systemic treatment alone (240 patients) or locoregional treatment (265 patients). All patients in the second group also received systemic treatment and were divided into two other groups: before or after locoregional treatment. Sixty-eight percent of patients in the locoregional treatment group also received local radiotherapy after surgery. The results showed a benefit for the group of patients treated with locoregional treatment with a 5-year OS rate of 72% vs. 33% in the systemic treatment group (HR 0.40, 95% CI 0.30–0.54, *p* < 0.0001). At least two considerations limit the applicability of these results to clinical practice; first, no patients received dual anti-HER2 blockade for HER2-positive tumors or CDK 4/6 inhibitors for luminal tumors, and second, the groups were not well-balanced with respect to age or tumor size. In fact, more patients with T3 tumors and younger patients were treated with systemic therapy before surgery [16].

Currently, there is still no consensus on whether or not to perform primary surgery. What is clear is the obligation to make an adequate selection of the patient before performing surgery on the primary tumor. The benefit is likely to be greater for those patients with fewer metastases and controlled systemic disease. Therefore, we believe that primary surgery should be considered in patients with the de novo oligometastatic disease. Also, we must consider the time without systemic therapy that is required during surgery and its potential relationship with survival.

## 3. Metastases Treatment

Systemic strategies such as chemotherapy, hormone therapy, immunotherapy, and/or targeted therapy are the mainstay of treatment for patients with MBC. Although OS has increased dramatically in the last decade, most patients die as a result of breast cancer. Median OS in patients with metastatic breast cancer ranges from 18 to 60 months, but tumor subtype, response to therapy, and tumor burden are important factors that can influence prognosis [17,18]. Furthermore, more and more patients remain free of progressive disease for many years.

Patients with the oligometastatic disease are known to have a better prognosis and probably benefit more from the radical treatment of metastases [19]. As indicated above, the lack of a clear definition makes it difficult to unify criteria to analyze the current evidence. A patient with the oligometastatic disease must be studied carefully to evaluate a radical approach. We recommend thorough staging. PET computed tomography, brain MRI, and bone scan should be performed routinely before considering the best strategy for this type of patient. If possible, a biopsy of the metastatic disease should be performed to rule out potential discrepancies. The only exceptions should be bone disease.

Although it is impossible to know whether the oligometastatic disease is a curable disease or not, more and more patients achieve the status of long-term no-evidence of disease after systemic treatment. In a “real-world data” study, patients with HER2-positive tumors and only one metastatic site showed longer PFS (*p* < 0.0001) and OS (*p* = 0.004) when treated with trastuzumab- and pertuzumab-based treatment in the first-line setting [20].

The role of a radical approach to treat all metastases in patients with the oligometastatic disease has not been widely explored. Surgery for metastases or specific radiation therapy techniques, such as stereotactic body radiation therapy (SBRT), external-beam radiation therapy (EBRT), or radiofrequency ablation (RFA), may be considered. The absence of remarkable biomarkers, the difficulty in designing the appropriate clinical trials, and the failure to offer this group of patients radical approaches in advanced-stage clinical trials are just some of the current problems that we face in treating patients with oligometastatic breast cancer.

The recent prospective phase IIR/III NRG-BR002 study randomized breast cancer patients with oligometastatic disease (≤4 metastases with ≤5 cm of diameter) and no progressive disease in the last 12 months after initiation of systemic disease to two arms; one of them received “standard of care” (SOC) with radiotherapy indicated only with palliative intent (arm 1) and the other was treated with total ablation of the lesions with SBRT or radiosurgery (arm 2). All breast cancer subtypes were allowed, but the vast majority of patients had ER+/HER2- tumors (79%). This trial failed to show a benefit in PFS (23 months in arm 1 vs. 19.5 months in arm 2) [21]. Despite these results, the lack of sufficient prospective data in this area makes it necessary to individualize the optimal approach according to each patient.

Below, we summarize the available data according to the location of the metastases amenable to radical treatment.

Patients with brain metastases without extracranial disease:

Brain metastases are a poor prognostic factor in different types of different cancer. In selected breast cancer patients, brain metastases may occur without evidence of extracranial disease. Although the reason for this phenomenon is unknown, the potential difficulty for some drugs to penetrate the central nervous system could be one of the reasons.

Local treatment of isolated brain metastases should be considered. For solitary lesions, surgical resection may be considered the preferred option [22,23]. However, many other aspects must also be considered, such as the performance status of the patient. In patients with more brain lesions, stereotactic radiosurgery (SRS) is recommended [24]. Whole brain radiotherapy (WBRT) could be used for patients in whom surgery or SRS is not possible due to the extent of brain disease [25]. In several studies, WBRT after surgery has shown a benefit in local recurrence but not in overall survival and was associated with neurological toxicity [26]. After local treatment, there is insufficient evidence for continuing systemic treatment or the duration if it is administered.

Patients with isolated liver involvement:

Liver metastases, together with lung and bone metastases, are the most frequent location of metastases in breast cancer. The 5-year survival rate for these patients is 4–12% [27]. Approximately 5–15% of patients have isolated or oligometastatic disease in the liver [28]. Although no prospective clinical trials have been conducted to explore the potential benefit of hepatic metastasectomy, retrospective data have shown increased survival in selected patients with a 5-year survival rate greater than 30% in most studies and median survival between 25 and 70 months [29]. Resection with clear margins (R0), younger patients, hormone receptor-positive or HER2-positive tumors, and those patients with an initial response to therapy were associated with better outcomes [30,31,32,33,34]. In a European matched case-control study, the combination of systemic therapy (including trastuzumab for HER2-positive tumors) with surgery for liver metastases showed benefit compared with systemic therapy alone (82 vs. 31 months, HR 0.28; *p* < 0.001) [35]. Subsequent liver resection has not been adequately evaluated but may be considered in selected patients [36].

If surgery cannot be performed due to patient performance status or deep lesions, other approaches, such as local ablation with radiofrequency (RFA) or microwave ablation (MWA, have also shown benefit in retrospective studies. Laparoscopic RFA without surgery has shown long-term survival with better OS for the RFA arm compared with systemic treatment (47 months vs. 9 months *p* = 0.0001) [37].

Patients with lung metastases:

Isolated lung metastases should be managed appropriately. Between 7–66% of all lung lesions that undergo surgery are not breast metastases. Among them, primary lung cancer is of special interest because proper diagnosis and treatment can increase the chance of cure [2].

There are no randomized trials comparing local treatment versus systemic treatment alone, and the results of retrospective studies are unclear. Although some benefit has been reported in different studies, there is significant variation in survival rates. One of the most important studies showed a 5-year survival rate of 38% with a median OS of 37 months for patients with complete lung resection. Some prognostic factors seem to be associated with a better outcome, such as disease-free interval (DFI) >36 months, solitary lung metastases, ER-positive tumors, small lesion size, and complete resection without positive margins [38]. In a meta-analysis of 16 studies, the 5-year survival rate after pulmonary resection was 46%. In this study, a DFI < 3 months, >1 metastatic lesion and an ER-negative tumor were poor prognostic factors [39]. The morbidity and mortality associated with this pulmonary surgery must be considered, so it is essential to select who can benefit from it.

Like other localizations of metastases, radiotherapy techniques, such as EBRT/SBRT, could be considered with similar results in survival rates compared to surgery with good local control and low comorbidity. The randomized phase II SABR-COMET trial compared SBRT and palliative standard of care versus palliative standard of care alone in various cancer types. Fifty percent of the metastatic lesions were lung metastases. There was a better OS when SBRT was administered, with a median OS of 50 months vs. 28 months (HR 0.47, 95% CI 0.27–0.81). The upper limit of benefit in this study was 5 metastatic lesions [40].

A recent meta-analysis of 10 studies (prospective and retrospective) comprising 467 patients with oligometastatic breast cancer treated with SBRT showed a 1- and 2-year local control rate of 97% (95% CI 95–99%) and 90% (95% CI 84–94%) respectively, with an OS of 93% at 1 year and 81% at 2 years [41].

Patients with bone metastases:

Bone is the most common site of breast metastases. Local treatments are usually indicated in the case of spinal cord compression, pathologic fracture, or pain.

Local treatment with “curative” intent could be offered for selected patients with the oligometastatic disease and few bone lesions. Retrospective data showed better symptom control in these patients and some survival benefits [42]. However, to make the best decision, it must be considered that these tumors usually have a better prognosis, and a good response to systemic treatment and bone surgery can entail significant morbidity. Radiotherapy for the oligometastatic disease might be a better option for these patients [40]. For patients with bone metastases, retrospective data have shown the benefit of hypofractionated stereotactic radiation therapy (HSRT) in those patients with bone-only oligometastases (5-year and 10-year OS rates of 83% vs. 75%) versus those patients with no bone-only disease (5-year and 10-year OS rates of 31% and 17%; *p* = 0.002) [43]. For isolated sternal metastases, surgery or radiotherapy should be indicated [44,45]. A phase 2 trial evaluated the efficacy of treatment with SBRT and fractionated intensity-modulated radiotherapy (IMRT) in all metastases of 54 patients with 92 metastatic lesions (most of them bone metastases). Results of this study showed a 1- and 2-year PFS of 75% and 53%, respectively [46].

## 4. Tools for Early Diagnosis of Oligometastatic Disease

More accurate diagnostic tests are needed to diagnose a larger number of patients in an oligometastatic setting. From an imaging point of view, PET-CT seems to be the most sensitive tool. However, its use in the early breast cancer field is commonly restricted to patients with equivocal metastatic disease. It is currently unknown whether the routine use of PET-CT is valid for identifying more patients with oligometastatic disease. In the PHERGain study, patients with HER2-positive early breast cancer were treated with a chemotherapy-free regimen [47]. However, PET-CT was routinely performed to rule out subclinical metastases. These patients were treated with double anti-HER2 therapy in combination with chemotherapy. Surgery for primary cancer was discussed on a case-by-case basis. The results of this study will help in part to address the role of breast cancer staging using PET-CT.

From the “molecular” point of view, the expanding use of liquid biopsy is opening a new area of opportunities. Although the role of circulating tumor DNA (ctDNA) is still under investigation, it seems clear that the detection of ctDNA after local treatment of early breast cancer is associated with an increased risk of relapse [48]. Furthermore, in these patients, the median time to detect macroscopic metastases is usually less than 1 year. For this reason, early detection of a positive ctDNA analysis could help to diagnose metastatic lesions earlier (potentially oligometastatic disease). Thus, close radiological follow-up for those patients with the presence of ctDNA in liquid biopsy might be of interest if future clinical trials help us to clarify the potential advantages of early vs. late metastatic disease diagnosis. On the other hand, in those patients with the oligometastatic disease who have been treated with a radical approach, there are doubts about the best “adjuvant” treatment. If the presence or not of a positive ctDNA analysis could lead to the need for further systemic treatment should be explored as well [49,50].

Trials addressing therapeutic strategies in patients without metastases but with “molecular oligometastatic disease” have just begun.

## 5. Conclusions and Future Perspectives

Although MBC remains an incurable disease, there are patients with a limited number of metastatic lesions that, in addition to systemic therapy, can be treated with “radical therapy” and sometimes reach the status of long-term no-evidence of disease. Whether or not these patients can be considered cured is still a matter of debate. Dramatic improvements in systemic therapies, including anti-HER2 agents for HER2-positive diseases, Cyclin-Dependent Kinases 4/6 inhibitors for luminal tumors, or immunotherapy and antibody-drug conjugates in triple-negative breast cancer, all showing important increases in OS, along with the most sensitive evaluations to diagnose metastatic disease earlier, are important components in the management of patients with oligometastatic breast cancer.

Although the optimal time to proceed with radical management of the oligometastatic disease is currently unknown, it seems reasonable to consider this approach during first-line therapy and, if possible, when the tumor has shrunk. The multidisciplinary tumor board is essential here to guarantee the best chances of success.

The lack of well-designed clinical trials in patients with oligometastatic disease makes it difficult to know what the optimal management is. This is why we urgently need to design strategic clinical studies for these patients. As we previously published [1], the limitations of current trials (which include a very heterogeneous population) lead us to propose academic research strategies in patients with selected breast cancer subtypes.

In the meantime, our opinion is to treat these patients with a “neoadjuvant-like” systemic therapy followed by the radical procedure (either with surgery and/or radiotherapy) followed by adjuvant systemic treatment. Although we cannot guarantee a cure for these patients at this time, many of them will remain without evidence of disease for many years, perhaps throughout life. It is our obligation as physicians to do our best for them.

## Figures and Tables

**Figure 1 cancers-14-05229-f001:**
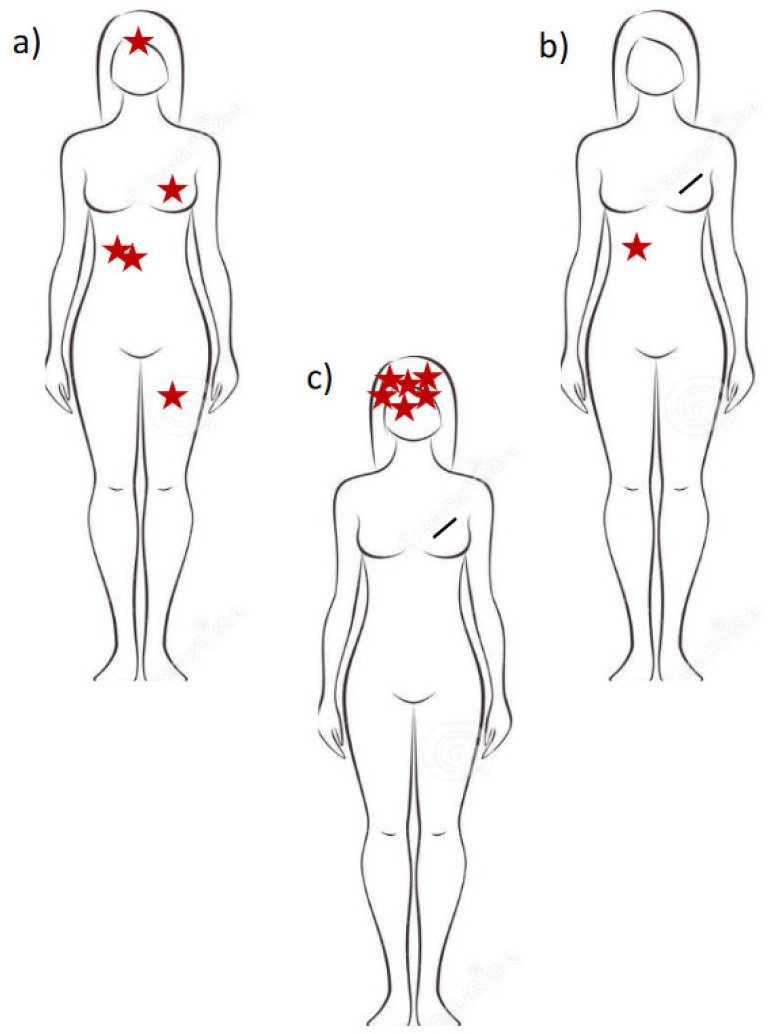
Oligometastatic disease. Model of different presentations of oligometastatic breast cancer: (**a**) Patient with the de novo oligometastatic disease: primary breast cancer, and two metastases in the liver, one in the bones and one in the brain; (**b**) Patient with an isolated liver metastasis; (**c**) Patients with six brain lesions: a radical approach could potentially be performed on all these metastases.

## Data Availability

No new data were created or analyzed in this study. Data sharing is not applicable to this article.

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
