# Peer review of "Oligometastatic Disease: When Stage IV Breast Cancer Could Be “Cured”"

_cancers, 2022, doi:10.3390/cancers14215229_

Round 1

Reviewer 1 Report

The authors have compiled an overall excellent review covering the most important studies completed in the oligometastatic space of breast cancer which is a rapidly evolving and exciting field. There are minor typographical errors which need corrected (Eg PET-TC written instead on PET-CT in the paragraph of "Tools of early diagnosis of oligometastatic disease"). 

Agree with the authors that despite conflicting data from multiple trials, radical locoregional treatment for patients with oligometastatic disease should be individualized on a case-by-case basis ideally with the support of a multidisciplinary tumor board. More prospective clinical trials specifically designed for oligometastatic disease in the era of modern systemic therapy are definitely needed.

Reviewer 2 Report

Dear Authors, 

Thank you for your submission. Your paper provides a clear, well-written narrative review of an unsolved problem in breast cancer management. The major issue of the manuscript is its lack in originality, as many similar reviews have been published, some in recent times. This problem could be overcome by a major revision that should aim to enhance the originality of the paper without sacrificing its readability and clearness. In general, I feel the manuscript touches lightly upon several interesting themes, but never delves in deep enough. While a full account of what is going on in every aspect of the oligometastatic disease might be overwhelming and transcend the purpose of this review, a little detail in some aspects may make a more significant addition to the existing literature (i.e. a wider approach on the topic of liquid biopsies). 

Citations are adequate, albeit a little few for a comprehensive review such as is proposed. Some papers that might be of interest:

Viani GA, Gouveia AG, Louie AV, Korzeniowski M, Pavoni JF, Hamamura AC, Moraes FY. Stereotactic body radiotherapy to treat breast cancer oligometastases: A systematic review with meta-analysis. Radiother Oncol. 2021 Nov;164:245-250. doi: 10.1016/j.radonc.2021.09.031. Epub 2021 Oct 6. PMID: 34624408.

Trovo M, Furlan C, Polesel J, Fiorica F, Arcangeli S, Giaj-Levra N, Alongi F, Del Conte A, Militello L, Muraro E, Martorelli D, Spazzapan S, Berretta M. Radical radiation therapy for oligometastatic breast cancer: Results of a prospective phase II trial. Radiother Oncol. 2018 Jan;126(1):177-180. doi: 10.1016/j.radonc.2017.08.032. Epub 2017 Sep 21. PMID: 28943046.

The only figure present is well-drawn and clear, although its general efficacy is debatable. A table of enrolled studies might perhaps aid comprehension further.

Language is fluent, and only a few typos are present within the manuscript.

Overall, the main concern remains originality. A new perspective on the subject may perhaps add some value to the work. 

Round 2

Reviewer 2 Report

Thank you Authors for addressing the manuscript as suggested. Though it is not the first manuscript of this kind in the existing literature, I now feel it is particularly comprehensive and brings a special clarity in respect to other published papers.